# Functional Crosstalk between PCSK9 Internalization and Pro-Inflammatory Activation in Human Macrophages: Role of Reactive Oxygen Species Release

**DOI:** 10.3390/ijms23169114

**Published:** 2022-08-14

**Authors:** Rafael I. Jaén, Adrián Povo-Retana, César Rosales-Mendoza, Patricia Capillas-Herrero, Sergio Sánchez-García, Paloma Martín-Sanz, Marina Mojena, Patricia Prieto, Lisardo Boscá

**Affiliations:** 1Instituto de Investigaciones Biomédicas Alberto Sols, CSIC-UAM, 28029 Madrid, Spain; 2Centro de Investigación Biomédica en Red de Enfermedades Hepáticas y Digestivas (CIBEREHD), Instituto de Salud Carlos III, 28029 Madrid, Spain; 3Departamento de Farmacología, Farmacognosia y Botánica, Facultad de Farmacia, Universidad Complutense de Madrid, 28040 Madrid, Spain; 4Centro de Investigación Biomédica en Red de Enfermedades Cardiovasculares (CIBERCV), Instituto de Salud Carlos III, 28029 Madrid, Spain

**Keywords:** PCSK9, LDL, ROS, atherosclerosis, macrophage, cholesterol, TLR4

## Abstract

Atherosclerosis is a cardiovascular disease caused mainly by dyslipidemia and is characterized by the formation of an atheroma plaque and chronic inflammation. Proprotein convertase subtilisin/kexin type 9 (PCSK9) is a protease that induces the degradation of the LDL receptor (LDLR), which contributes to increased levels of LDL cholesterol and the progress of atherosclerosis. Given that macrophages are relevant components of the lipidic and inflammatory environment of atherosclerosis, we studied the effects of PCSK9 treatment on human macrophages. Our data show that human macrophages do not express PCSK9 but rapidly incorporate the circulating protein through the LDLR and also activate the pro-inflammatory TLR4 pathway. Both LDLR and TLR4 are internalized after incubation of macrophages with exogenous PCSK9. PCSK9 uptake increases the production of reactive oxygen species and reduces the expression of genes involved in lipid metabolism and cholesterol efflux, while enhancing the production of pro-inflammatory cytokines through a TLR4-dependent mechanism. Under these conditions, the viability of macrophages is compromised, leading to increased cell death. These results provide novel insights into the role of PCSK9 in the crosstalk of lipids and cholesterol metabolism through the LDLR and on the pro-inflammatory activation of macrophages through TLR4 signaling. These pathways are relevant in the outcome of atherosclerosis and highlight the relevance of PCSK9 as a therapeutic target for the treatment of cardiovascular diseases.

## 1. Introduction

Proprotein convertase subtilisin/kexin type 9 (PCSK9) plays a major role in cholesterol metabolism and atherosclerosis by modulating the circulating levels of low-density lipoprotein (LDL), the key molecule that transports cholesterol in the blood [1,2]. PCSK9 interacts with the epidermal growth factor repeat A domain of the LDL receptor (LDLR; [3]) at the cell surface of hepatocytes, lymphocytes, macrophages (Mφ), endothelial cells, and vascular smooth muscle cells, inducing their combined internalization through clathrin-dependent endocytosis and lysosomal degradation, preventing LDLR recycling [4,5]. As a result, LDLR expression on the cell membrane decreases, causing LDL to accumulate in the subendothelial layer and be retained within the proteoglycan matrix [6]. LDL tends to be oxidized by reactive oxygen species (ROS) released by surrounding cells, generating oxidized LDL (oxLDL) molecules that exacerbate the inflammatory response by interacting with scavenger receptors, such as CD36, and toll-like receptors, such as the toll-like receptor TLR4 [7,8,9].

TLR4 belongs to the family of pattern-recognition receptors. These receptors identify specific pathogen-associated and damage-associated molecular patterns (PAMPs and DAMPs, respectively) that activate the innate immune response in the host and enhance the inflammatory reaction [10,11,12,13]. TLR4 is widely expressed in immune cells such as Mφ and other cell types, such as adipocytes, cardiomyocytes, and muscle cells. It is known to be a promiscuous receptor with a wide range of ligands, including the canonical ligand bacterial lipopolysaccharide (LPS) as well as other non-canonical effectors [14,15,16]. This allows TLR4 to respond to different stimuli and rapidly trigger an inflammatory reaction to fight against the damaging agent. TLR4 challenge activates the TRIF/IRF3-dependent pathway that promotes IRF3-dependent transcription [17] and MyD88-dependent signaling, which induces IKK activation, followed by phosphorylation and ubiquitin-dependent degradation of the main inhibitor of NF-κB, IκB-α, thus promoting NF-κB nuclear translocation and regulation of transcription of more than 1000 genes in Mφ, implicated in inflammation, apoptosis, proliferation, and oxidative stress [15,18,19].

Regarding apoptosis, TLR4 can regulate caspase-3-mediated apoptosis/pyroptosis [20] and contributes to NLRP3 inflammasome assembly, a multimeric protein complex that mediates caspase-1 cleavage, which in turn promotes IL-1β pro-inflammatory cytokine activation and release [21,22,23]. Although TLR4 function is crucial for triggering the immune response [10,24,25], uncontrolled TLR4 activation could cause excessive tissue damage, leading to chronic inflammation. Indeed, sustained activation of the TLR4 pathway in the vascular system promotes atherosclerosis progression by inducing foam cell formation, lipid accumulation, and even atheroma plaque rupture [11,12,25]. These detrimental effects are mediated, at least in part, by excessive NLRP3 activation and IL-1β release, which exacerbates the inflammatory reaction, and by increasing Mφ apoptosis, which favors the establishment of the necrotic core in later stages of the disease [26,27].

Although the role of PCSK9 in atherosclerosis and LDLR degradation is well known, the ability of PCSK9 to modulate other responses is not fully understood. Interestingly, previous results have shown that PCSK9 is internalized in LDLR^−/−^ Mφ, suggesting that different receptors may interact with PCSK9 and promote its endocytosis [28,29]. Our results confirm that human macrophages (hMφ) do not show a significant (if any) expression of PCSK9. Moreover, in addition to LDLR-dependent incorporation of extracellular PCSK9, this protease facilitates the rapid activation of TLR4, which in turn induces an enhancement in ROS production, leading to moderate apoptosis and contributing to a significant pro-inflammatory polarization, including the release of TNF-α and the processing of pro-IL-1β. In addition to this, the incorporation of PCSK9 in hMφ reduces the expression of genes involved in lipid metabolism and the efflux of cholesterol, as reflected by the reduction in the mRNA levels of the ATP-binding cassette transporter ABCA1.

## 2. Results

### 2.1. Incorporation of PCSK9 in Human Macrophages Enhances ROS Synthesis and Reduces Cell Viability

The circulating serum levels of PCSK9 vary between healthy donors, which is relevant in terms of incorporation into LDLR expressing cells, including monocytes/macrophages (Figure 1A). Moreover, this individual variation results in different levels of PCSK9 incorporation in isolated human monocytes (Figure 1B). However, after differentiation of human monocytes into macrophages (hMφ), the presence of PCSK9 is not detected but it can be rapidly incorporated from an external source of protein; in this case, PCSK9 incorporation occurs after hMφ incubation with increasing concentrations of culture medium from HepG2 cells, a human hepatic cell line that expresses and releases PCSK9 to the culture medium (Figure 1C). To assess the ability of hMφ to express *PCSK9*, these cells were polarized into pro-inflammatory (M1-like for short) and alternatively activated (M2-like) profiles as previously described [30] and the mRNA levels of *PCSK9* were determined using HepG2 cells as controls. As Figure 1D,E shows, hMφ were unable to express this mRNA regardless of polarization phenotype and the time after incubation with exogenous PCSK9. The kinetics of PCSK9 incorporation into hMφ was determined after incubation of polarized cells in a medium containing 0.5 µg/mL of recombinant PCSK9 (Figure 1F), showing the highest incorporation into M2-like hMφ cells. This lack of expression of *PCSK9* by hMφ was also observed in the murine counterparts. As Appendix A shows, PCSK9 was present in several mice tissues but, at the mRNA level, peritoneal Mφ failed to show significant levels of *Pcsk9* (Appendix A).

This incorporation of PCSK9 into hMφ enhanced the synthesis of ROS in a dose-dependent manner (Figure 2A). Interestingly, and because ROS production is associated with pro-inflammatory polarization, inhibition of TLR4 activity with TAK242 reduced ROS synthesis to near-basal levels. Furthermore, treatment of hMφ with exogenous PCSK9 enhanced the percentage of annexin V^+^-cells, a process that was reduced after inhibition of TLR4 activity and, to a lower extent, after *LDLR* silencing (Figure 2B). Treatment of hMφ with heat-inactivated PCSK9 (denatured) failed to induce apoptosis and ROS synthesis. This PCSK9-dependent ROS increase was also observed after functional polarization of hMφ and was partially reduced after silencing *LDLR* in these cells (Figure 2C). Indeed, analysis of the incorporation process of PCSK9 in hMφ showed a mobilization of LDLR and TLR4, suggesting a potential crosstalk between both receptors (Figure 2D,E).

### 2.2. Incorporation of PCSK9 in Human Macrophages Reduces LDLR Content and Decreases the mRNA Levels of Genes Involved in Lipid and Cholesterol Metabolism

The incorporation of PCSK9 in hMφ is mainly mediated after interaction with the LDLR on the cell surface, promoting its internalization (Figure 2D) and subsequent lysosomal degradation, as described by several groups [5,14]. This degradation of the LDLR was accompanied by a significant decrease in the corresponding *LDLR* mRNA levels (Figure 3A) as well as in other genes involved in lipid metabolism, such as *SREBF1*, coding for the sterol regulatory element binding transcription factor 1 (SREBF-1), which has been described to be profoundly associated with foam cell formation [31]. SREBF-1 is a transcription factor that regulates the expression of genes implicated in lipid synthesis, such as fatty acid synthase (*FASN*) and 3-hydroxy-3-methylglutaryl-CoA reductase (*HMGCR*), involved in fatty acid and cholesterol synthesis, respectively [31]. Accordingly, *FASN* and *HMGCR* levels decrease after incubation of hMφ with PCSK9 (Figure 3A).

We also analyzed the effect of PCSK9 on the mRNA levels of *ABCA1* and *ABCG1*, which encode for proteins of the superfamily of ATP-binding cassette (ABC) transporters responsible for carrying excess cholesterol and phospholipids from lysosomes and endosomes to the cell membrane for its export [32,33,34]. Interestingly, while *ABCA1* levels decreased (the transporter that participates in cholesterol and phospholipids efflux onto lipid-poor ApoA-I and nascent HDL), the levels of *ABCG1* (with a broader selectivity for phospholipids, cholesterol, and oxidized cholesterol export) increased significantly. These data suggest that PCSK9, in addition to binding LDLR for its internalization in hMφ, modifies the expression pattern of genes involved in lipid and cholesterol metabolism. To assess whether the biosynthesis of cholesterol also regulates the expression levels of *LDLR*, we incubated HepG2 cells (as cells expressing PCSK9) and hMφ with the squalene synthase inhibitor zaragozic acid A [35]. As Figure 3B,C shows, the *LDLR* mRNA levels remained unchanged in HepG2 cells; however, incubation for 24 h of hMφ with zaragozic acid A increased *LDLR* levels but did not affect the reduction in *LDLR* levels following PCSK9 addition. Furthermore, this increase in LDLR induced after treatment with zaragozic acid A was accompanied by an enhancement in ROS production and annexin V^+^-cells (Figure 3D,E). These results suggest that, in the presence of PCSK9, the levels of LDLR contribute to ROS synthesis and cell viability of hMφ.

### 2.3. PCSK9 Activates Pro-Inflammatory and Pro-Apoptotic Signaling Pathways in Human Macrophages

As shown in Figure 2A,B,E and confirming previous reports [16,36], the incorporation of PCSK9 in hMφ activates the NF-κB signaling pathway, manifesting its pro-inflammatory role. Consistent with this fact, treatment of hMφ with PCSK9 induced the expression of pro-inflammatory genes, such as *TNF* and *IL1B*, but also led to an increased expression of the inflammasome gene *NLRP3* (Figure 4A). Interestingly, other genes, such as *ITGAM* (encoding the CD11b macrophage receptor) or *CYBB* (encoding the NOX2 protein that is involved in ROS production), remained unchanged or were even significantly decreased, as was the case of the *CD36* scavenger receptor (Figure 4A). Furthermore, consistent with the drop in *CD36* levels, a receptor involved in the uptake of oxLDL, a decrease in the labeling of CD36 at the cell plasma membrane was observed in hMφ treated with PCSK9 for 18 h (Figure 4B). In addition to this, the levels of the mannose receptor CD206 were decreased in these hMφ, while CMXROS labeling, an indicator of the mitochondrial inner membrane potential, was decreased, suggesting a pro-apoptotic profile (Figure 4B).

To characterize this emerging pro-inflammatory profile after PCSK9 incorporation into hMφ, the accumulation in the culture medium of IL-1β and TNF-α was determined. As Figure 4C shows, a significant increase in these pro-inflammatory cytokines was observed. Interestingly, inhibition of TLR4 activity with TAK242 significantly reduced the accumulation of IL-1β and TNF-α, suggesting that TLR4 activity is linked to the appearance of the pro-inflammatory profile involved in the synthesis of these cytokines. The release of IL-1β into the culture medium requires proteolytic processing driven by the p10 form coming from cleaved-caspase-1. Cleavage of pro-caspase-1 is observed after activation of the NLRP3 inflammasome [37,38]. In agreement with this activation, the levels of the p10 form of caspase-1 significantly increased after PCSK9 incubation of hMφ (Figure 4D).

To further confirm this link between the LDLR and TLR4 activation after incubation of Mφ with PCSK9, we measured the synthesis of ROS and the mRNA levels of *Il1b* and *Tnf* in peritoneal Mφ from wild-type and either *Ldlr*^−/−^ or *Tlr4*^−/−^ mice. As Appendix A shows, a significant reduction in ROS production was observed in the absence of *Ldlr* or *Tlr4*; however, the expression levels of *Tnf* and *Il1b* and the accumulation of these pro-inflammatory cytokines in the culture medium failed to show these differences in the absence of *Ldlr* (Appendix A).

### 2.4. The Pro-Inflammatory and Pro-Apoptotic Effects after Incorporation of PCSK9 in Human Macrophages Are TLR4 Dependent

Incubation of hMφ with PCSK9 promotes the appearance of annexin V^+^-cells in an LDLR- and TLR4-dependent manner (Figure 2B). This induction of apoptotic cells was evaluated following the activation of caspase-3 (cleaved form) that was evident after 1 h of treatment with PCSK9 (Figure 5). In addition to this, the activation of p38, a mitogen-activated protein kinase (MAPK) implicated in pro-inflammatory signaling and involved in TNF-α and IL-1β production [1,6,11,36,39], reached a maximum at 2 h after incubation of the cells with PCSK9 (Figure 5). The activation of the TLR4-dependent NF-κB pathway was confirmed by the rapid increase in the phosphorylation of IκB-α and p65 and by the inhibitory effect that pre-incubation with the TLR4 inhibitor TAK242 has on the processing of caspase-3 and the phosphorylation of p38 MAPK, IκB-α, and p65, reinforcing the pro-inflammatory activation switch mediated by TLR4 (Figure 5).

## 3. Discussion

In the present work, we analyze the effects of PCSK9 on hMφ, including its ability to modulate the polarization of these cells. This is in addition to the effects on LDLR degradation [40], thus underscoring its deleterious role in atherosclerosis and encouraging the design of future PCSK9-targeted therapies [41], in addition to its role in other pathologies, from Alzheimer’s to sepsis [42,43]. Furthermore, a potential positive role for PCSK9 in other pathological processes that require a pro-inflammatory profile of Mφ, as occurs in cancer, cannot be disregarded [44,45].

Beginning with atherosclerosis as a chronic inflammatory disorder of the blood vessels characterized by the accumulation of lipids in the arterial wall, forming an atherosclerotic plaque that restricts blood flow, it has been described that circulating PCSK9 may contribute to several cardiovascular diseases, such as coronary artery disease and myocardial infarction [12,41,46,47]. Therefore, an understanding of the molecular mechanisms involved in atherosclerosis progression is required for its prevention and the design of efficient therapeutic approaches against atherothrombotic pathologies. Since PCSK9 induces LDLR degradation, this favors an increase in circulating cholesterol levels. In this context, reduction of PCSK9 levels has become a promising therapeutic option. However, recent research has proposed that PCSK9 also modulates the activity of other receptors, such as those that recognize oxLDL and the TLRs, including TLR4, in different cell types [12,41,48]. In this sense, and because Mφ are key players in atherosclerosis progression, the knowledge of PCSK9-dependent effects on Mφ is of paramount importance. Indeed, increased lipid uptake by vascular Mφ transforms them into foam cells, which eventually undergo apoptosis and necrosis, which promotes the progression of atherosclerosis [2,39].

First, we observed that hMφ did not express PCSK9 but exhibited a rapid uptake of the exogenous protein that was almost maximal after 5 min of addition, followed by LDLR degradation at 30 min, which is consistent with previous studies [39,49]. Interestingly, incubation of hMφ with PCSK9 failed to induce its expression in these cells. Furthermore, PCSK9 treatment of hMφ downregulated the levels of genes involved in lipid metabolism, particularly those that modulate cholesterol synthesis (*SREBP1*, *FASN*, and *HMGCR*) and efflux (*ABCA1*), but not *ABCG1*, which is more abundant in the membranes of the endoplasmic reticulum, contributing to phospholipid mobilization [50]. This downregulation of *ABCA1* mRNA levels impairs the cholesterol efflux favoring lipid aggregation [51]. Since *PCSK9* expression is also regulated by the transcription factor SREBP-1 in various cell types, our results indicate that PCSK9 can initiate negative intra- and inter-cellular feedback that results in the downregulation of the transcription of genes dependent on the SREBP-1 pathway [31,52,53,54]. SREBP-1 is under the control of SREBP-2 and, in macrophages, SREBP-1 regulates the expression of scavenger receptors involved in efferocytosis and phagocytosis, linking lipid metabolism and inflammatory responses [53]. Indeed, crosstalk between the LXR and SREBP-1 transcription factors is relevant in the regulation of cholesterol metabolism and efflux in macrophages [52,53]. Therefore, the ability of PCSK9 to modulate SREBP-1 and its targets can contribute to lipid metabolism dysregulation [55]. Together, these results indicate that PCSK9 uptake causes an alteration in the lipid profile in hMφ that is involved in foam cell formation. This effect appears to be dependent on LDLR, as described in murine Mφ [1].

Our data show that PCSK9 is a relevant inducer of the pro-inflammatory response in hMφ, at least in part by activating TLR4, via the canonical NF-κB pathway, which explains the increased TNF-α and IL-1β expression, the NLRP3 inflammasome activation, and the induction of caspase-3-mediated apoptosis [41,56]. This profile was also observed in hMφ silenced for *LDLR*, as shown by the production of ROS, as well as in Mφ from *Ldlr*^−/−^ mice. At the same time, overexpression of *LDLR* after treatment with zaragozic acid A, in addition to elevating *LDLR* levels, enhanced ROS production and increased the presence of annexin V^+^ cells after incubation with PCSK9, suggesting a connection between the extent of LDLR internalization and oxidative stress. However, additional crosstalk between the internalization of the PCSK9–LDLR complex and TLR4 cannot be fully excluded, as has been suggested by clinical evidence [6]. TLR4-dependent NF-κB activation may be sustained since we observed increased phosphorylation of p38 for longer periods, which also acts as an activator of this pathway [57]. Interestingly, the phosphorylation of IκB-α was rapid, whereas the phosphorylation of p65 at Ser536 took longer periods. In general, p65 phosphorylation is conceived as a precondition for NF-κB transcriptional activity; however, it has been also proposed that sustained *p*-p65 attenuates NF-κB signaling and avoids excessive inflammation [58].

The pro-inflammatory cytokines TNF-α and IL-1β are known to contribute to the development of atherosclerosis and foam cell formation [59], further evidencing the detrimental role of PCSK9 in this pathology. Interestingly, both cytokines were described to impair proper lipid turnover in Mφ, partially explaining the genetic alteration in lipid metabolism that we observed earlier [59]. The NLRP3 inflammasome has recently gained attention in atherosclerosis research since crystalline cholesterol and oxLDL were shown to activate this complex [21]. Our data suggest that PCSK9 acts as an activator of NLRP3, revealing a new mechanism for NLRP3 involvement in atherosclerosis and underlining the importance of the inflammasome in this pathology.

By inhibiting TLR4 signaling with the inhibitor TAK242, we observed that the activation of the NF-κB pathway by PCSK9, in particular, the release of pro-inflammatory cytokines (TNF-α and IL-1β), occurs via TLR4. In fact, TLR4 internalization after PCSK9 treatment occurs even before LDLR degradation, suggesting that TLR4 activation may be due to direct interaction either with PCSK9 on the cell surface or with the initial PCSK9-LDLR complex, instead of being induced by LDLR internalization and degradation. This is consistent with the preliminary results of our group, which showed that PCSK9 was internalized in LDLR^−/−^ Mφ (albeit with different kinetics), suggesting the existence of alternative PCSK9 internalization pathways. In this regard, Hapton et al. found that PCSK9 shares an identical domain with resistin, a well-known TLR4 ligand, further supporting our hypothesis [60]. Interestingly, activation of the MyD88-independent pathway, whose main mediator is IRF3, requires the endocytosis of TLR4 [61], suggesting that PCSK9 can also be internalized together with TLR4. Moreover, it has been shown that IRF3 activation can downregulate lipid metabolism gene expression [62] and activate both NLRP3 inflammasome and apoptosis pathways [63,64]. Overall, this creates a new paradigm in the comprehension of PCSK9 function in Mφ metabolism and atherosclerosis progression.

The ability of PCSK9 to modulate a wide range of pro-inflammatory responses could indicate that increased PCSK9 enhances the susceptibility to developing inflammatory diseases. The plasma levels of PCSK9 correlate with the probability of future adverse cardiovascular events and atherosclerosis progression [1,41,65]. This susceptibility could be detected in healthy individuals due to the great variability observed in PCSK9 plasma levels in healthy donors, depicting PCSK9 as a biomarker of cardiovascular risk.

From a therapeutic point of view, several PCSK9 inhibitors are available, mainly as monoclonal antibodies. They are prescribed for atherosclerotic cardiovascular diseases and patients with familial hypercholesterolemia, being efficient at lowering LDL cholesterol [66,67]. These inhibitors are administered via subcutaneous injections. Indeed, they represent a complementary option to statins, which remain a widely prescribed and secure medication to treat atherosclerosis [66]. In fact, statins increase the expression of the LDLR, although they favor the induction of the p38 and NLRP3 inflammasome pathways and increased PCSK9 levels [65,66,68]. Therefore, a potential additional approach could be a combinational therapy with statins and PCSK9 inhibitors.

In conclusion, the present work shows an additional mechanism of action of PCSK9 in Mφ involving TLR4 activation, implicating the regulation of a plethora of responses, including lipid metabolism, inflammation, and apoptosis. All of these are detrimental to the progression of atherosclerosis. Taken together, our results demonstrate that, in addition to being a therapeutic target, PCSK9 can serve as a biomarker, underscoring the importance of PCSK9 inhibitors in the treatment of atherosclerosis.

## 4. Methods and Materials

### 4.1. Materials

Common reagents were from Sigma-Aldrich-Merck (Madrid, Spain) or Roche (Darmstadt, Germany). Murine and human cytokines were obtained from PeproTech (London, UK) or Merck. Antibodies were from Abcam (Cambridge, UK) or Cell signaling (Danvers, MA, USA). Specific *siRNA*s and a nonspecific scrambled *scRNA* control were from Dharmacon (Merck). Reagents for electrophoresis were from Bio-Rad (Madrid, Spain). Tissue culture dishes were from Falcon (Lincoln Park, NJ, USA), and serum and culture media were from Invitrogen (Life Technologies/Thermo-Fisher, Madrid, Spain).

### 4.2. Isolation of Human Monocytes and Preparation of Human Macrophages

Cells were prepared from buffy coats or fresh blood collected between 8 and 10 h from anonymous healthy donors who had fasted overnight, in agreement with Institutional and Centro de Transfusiones de la Comunidad de Madrid agreements (28504/000011), following previous protocols [69]. Donors were informed and they provided written consent following the ethical guidelines of the 1975 Declaration of Helsinki and the Committee for Human Subjects. To isolate human peripheral mononuclear cells (PBMC), buffy coats were treated with Ficoll (17-0300, Sigma-Aldrich-GE, Madrid, Spain) by carefully adding blood by soft dripping to prevent the two-phase mixture and then centrifuged for 25 min at 450× *g* at RT without brake. Plasma and PBMC fractions were collected from the upper-aqueous phase and washed twice with sterile PBS by centrifuging for 5 min at 300× *g* at RT. Remnant erythrocytes from the PBMC fraction were lysed by treating with NH_4_Cl solution (07850, Stem Cell, Saint Égrève, France), followed by washing with PBS. Cell count and viability were evaluated by flow cytometry (FACS-Canto II, Becton Dickinson, ref. 338962, Madrid, Spain) and trypan blue (T8154, Sigma). Finally, PBMC were centrifuged for 8 min at 300× *g* at RT, resuspended in FCS-free DMEM (11966-025, Gibco) with penicillin/streptomycin (15140/122, Gibco, Madrid, Spain), and seeded at 2 × 10^6^ cells/well in 6-well cell culture plates (353046, Falcon, Radnor, PA, USA). Human macrophages (hMφ) were prepared after culture in FCS-free DMEM for 1 h to induce monocyte cell adhesion. Then, 10% heat-inactivated FCS (10270/106, Gibco) was added to the cell media and left overnight. Cells were washed twice with sterile PBS to remove lymphocytes, and the culture medium was replaced with DMEM and 10% heat-inactivated FCS. Cells were incubated for 7 days, allowing human monocytes to differentiate into hMφ. After differentiation, the culture medium was replaced with RPMI1640 (21875, Gibco) and FCS 2% 18 h before the experiments.

### 4.3. Preparation of Elicited Peritoneal Mice Macrophages

C57BL/6J wild type (WT), *Tlr4*^−/−^ (029015 strain) and *Ldlr*^−/−^ (002207 strain) were from the Jackson Laboratory and bred in our animal facility. Animal experiments were approved by Institutional and administrative authorization (PROEX 228_17). Mice were maintained under controlled humidity and temperature in pathogen-free conditions and 12 h/12 h light/dark cycles. Male mice aged 9–12 weeks received an intraperitoneal injection of 2.5 mL of 3% (weight/vol) of thioglycollate broth [30]. Elicited peritoneal macrophages were prepared from light-ether anesthetized mice (4 animals per condition), killed by cervical dislocation and injected intraperitoneally with 10 mL of sterile RPMI1640 medium. The peritoneal fluid was carefully aspirated, avoiding hemorrhage, and kept at 4 °C to prevent the adhesion of the macrophages to the plastic. An aliquot of the cell suspension was used to determine the cell density in the peritoneal fluid. The cells were centrifuged for 10 min at 200× *g* at 4 °C, and the pellet was washed twice with 25 mL of ice-cold PBS. Cells were seeded at 1 × 10^6^/cm^2^ in RPMI1640 medium supplemented with 10% of heat-inactivated FCS and antibiotics. After incubation for 3 h at 37 °C in a 5% CO_2_ atmosphere, non-adherent cells were removed by extensive washing with PBS. Experiments were carried out in RPMI1640 medium and 2% of heat-inactivated FCS plus antibiotics.

### 4.4. Differentiation of Macrophages into Pro-Inflammatory and Alternatively Activated Cells

Cells were treated with 20 ng/mL of LPS and 10 ng/mL each of TNF-α, IL-1β, and IFN-γ for pro-inflammatory activation (M1 hMφ to abbreviate) or with 10 ng/mL each of IL-4, IL-10, and IL-13 for alternative activation (M2 hMφ) following previous protocols [30].

### 4.5. Cell Treatments

Monocytes or macrophages were incubated with human recombinant PCSK9 (CY-R2330, MBL Circulex) at the indicated times. Alternatively, supernatants from HepG2 cells, which release processed PCSK9 to the culture medium, were used. Heat-treated recombinant PCSK9 (80 °C for 10 min at 20 μg/mL in RPMI1640) was used as a negative control. TLR4 inhibition was achieved by treating cells with 1 µM TAK242 (CAS 243984-11-4, Merck) 30 min before PCSK9 addition. Incubation with zaragozic acid A (Z2626, Sigma-Aldrich) was carried out for 24 h before PCSK9 administration.

### 4.6. Human LDLR Silencing

The LDLR-specific *siRNA*s and a nonspecific control (*scRNA*) were from Dharmacon (DhamaFECT, Dharmacon ON-TARGETplus Smart pool *siRNA*; 3949, L011073-00-0005) and were used at 50 nM. Macrophages were transfected using lipofectamine RNAiMax reagent. At 48 h post-transfection, silencing *LDLR* resulted in 92 ± 4% decrease in *mRNA* levels and 88 ± 8% in total protein levels vs. the corresponding *scRNA*-treated cells (*n* = 4 different donors). Additionally, the levels of LDLR on the cell surface were negligible, as determined by immunofluorescence microscopy.

### 4.7. Measurement of ROS Production

ROS production was measured by incubating cells for 30 min at 37 °C in 5% CO_2_ in darkness with a 5 µM DCFH-DA fluorescent probe (2′-7′-dichlorofluorescein diacetate, D6683, Sigma). The oxidation of DCFH was quantified by flow cytometry. Briefly, cells were incubated with the indicated labeled antibodies and analyzed by flow cytometry (FACSCanto II, Beckton Dickinson, Madrid, ES, Spain) and positive cells were quantified using FlowJo software. DAPI (Life Technologies, Madrid, ES, Spain) was used to discriminate dead cells in the analysis. For cell counting, absolute counting beads were added (CountBright, Invitrogen, Madrid, Spain).

### 4.8. Measurement of Cell Membrane Receptors

The cell surface levels of the scavenger receptor CD36 (anti-human CD36-APC, 336208, Biolegend, San Diego, CA, USA) and the mannose receptor CD206 (anti-human CD206-PE, 321106, Biolegend) were determined by flow cytometry following previous protocols [70].

### 4.9. Evaluation of Mitochondrial Inner Membrane Potential

Mitochondrial membrane potential (ΔΨm) measurement in hMφ was monitored using 100 nM CMXROS (Red MitoTracker, M7512, Invitrogen). The fluorescent probe was incubated for 30 min at 37 °C in 5% CO_2_ in darkness following previous protocols [71].

### 4.10. Quantification of Annexin V^+^ Cells

Cells were harvested and washed in ice-cold PBS. After centrifugation at 4 °C for 5 min and 800× *g*, cells were resuspended in annexin V binding buffer (10 mM HEPES, pH 7.4, 140 mM NaCl, 2.5 mM CaCl_2_). Cells were labeled with annexin V–PE solution (Immunostep, Salamanca, Spain) and/or propidium iodide (PI, 100 μg/mL) for 15 min at RT in the dark. PI is impermeable to living and apoptotic cells but stains necrotic and apoptotic dying cells with impaired membrane integrity in contrast to annexin V, which stains early apoptotic cells. Macrophage viability was determined by flow cytometry in a BD-Canto II flow cytometer as previously described [30].

### 4.11. Protein Analysis by Western Blot

Cells were homogenized at 4 °C in a lysis buffer containing 10 mM Tris-HCl (pH 7.5), 1 mM MgCl_2_, 1 mM EGTA, 10% glycerol, 0.5% CHAPS (C3023, Sigma-Aldrich), and protease and phosphatase inhibitor cocktails (P8340, P5726, P0044, Sigma-Aldrich). Samples were vortexed for 20 min and centrifuged at 12,000× *g* for 15 min at 4 °C. Supernatants were stored at −20 °C. Protein concentration was determined by the Bradford assay (Bio-Rad). Equal amounts of protein (20–60 μg) from each fraction obtained were loaded into 8–12% of SDS-PAGE. Proteins were size fractionated, transferred to a PVDF membrane (Bio-Rad), and, after blocking with 5% of bovine serum albumin (BSA), incubated with the corresponding antibodies (Appendix A). Blots were developed by ECL protocol, and different exposition times were performed for each blot to ensure the linearity of the band intensities. Values of densitometry were determined using Image J version 1.52p software (NIH).

### 4.12. RNA Isolation and Analysis

RNA from cells was extracted in Trizol Ambion (AM9738, ThermoFisher, Madrid, Spain) following the manufacturer’s instructions. RNA was quantified in a NanoDrop 2000 spectrophotometer (ThermoFisher), and 1 µg of RNA was reverse-transcribed to cDNA with a Transcriptor First-Strand cDNA Synthesis kit (04379012001, Roche). qPCR assay was carried out with 5 µL of this template cDNA, 10 µL of the SYBR Green PCR Master Mix cocktail (4309155, ThermoFisher) and 250 nM forward and reverse primers (Appendix A). *RPLP0* (*36B4*) was chosen as a housekeeping endogenous control for normalization purposes. A qPCR reaction was carried out in MyIQ RealTime PCR System (BioRad, Madrid, Spain). Result analysis was conducted with the IQ5 program (BioRad) following the 2^−ΔΔCt^ method.

### 4.13. Immunofluorescence Analysis

After treatments, cells were fixed with 2% paraformaldehyde (Merck) for 20 min and then blocked and permeabilized with 2% BSA, 0.3% Triton X-100 (Sigma), and 5% normal goat serum (7481, Abcam) for 1 h. After being incubated with the corresponding primary antibody 1:100 at 4 °C overnight (Appendix A), the samples were incubated in the dark with secondary antibodies combined with Alexa Fluor 488 or 546 (Molecular Probes, Madrid, Spain) for 2 h 1:500 and then DAPI (D1306, Molecular Probes) 1:500 for 12 min, being gently washed with PBS between incubations. Coverslips were mounted in ProLong Gold 5 Antifade reagent (Invitrogen) and examined using a confocal spectral microscope TCS SP5 Leica.

### 4.14. Quantification of IL-1β and TNF-α

Cytokines were determined in cell culture supernatants using human and mouse specific kits, i.e., IL-1β ELISA Kits (900-T95 (PreproTech, London, UK) and RAB0275 (Merck)) and TNF-α ELISA Kits (900-T25 and 900-T54, PeproTech), following manufacturer’s instructions.

### 4.15. Statistical Analysis

Values in graphs correspond to the mean ± SD. The statistical significance of differences between the means was determined with GraphPad Prism 9.0.0. (GraphPad Prism 9 Software, San Diego, CA, USA) using a one-way analysis of variance (ANOVA) followed by Bonferroni post hoc test or Student’s *t*-test, as appropriate. A *p*-value < 0.05 was considered to be significant.

## 5. Conclusions

Human macrophages fail to express PCSK9, regardless of the functional polarization: PCSK9 incorporation in macrophages enhances ROS production and decreases cell viability.LDLR-dependent incorporation of PCSK9 decreases the expression of ABCA1.The pro-inflammatory activation after PCSK9 uptake is dependent on TLR4 signaling.

## Figures and Tables

**Figure 1 ijms-23-09114-f001:**
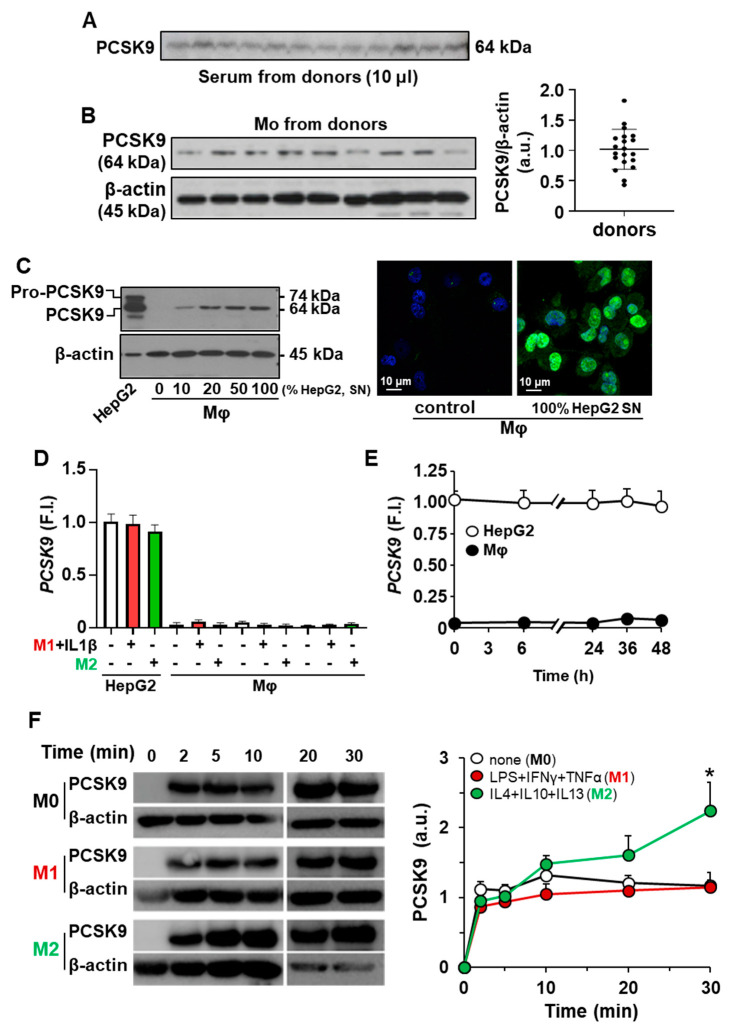
Incorporation of PCSK9 into human monocytes (Mo) and human macrophages (hMφ). (**A**) Quantification of circulating PCSK9 in human serum from healthy donors after overnight fasting. Identical amounts of serum were analyzed by Western blot. (**B**) Determination of PCSK9 in circulating Mo from healthy donors. (**C**) Incorporation of PCSK9 from the culture medium of HepG2 cells (SN, 48 h in culture) in hMφ differentiated from circulating Mo (**left panel**). Immunofluorescence of PCSK9 in hMφ (anti-PCSK9 antibody in green and DAPI in blue) treated for 30 min with a fresh medium or a conditioned medium from HepG2 cell cultures (**right panel**). (**D**) PCSK9 mRNA levels from HepG2 and human macrophages were treated for 18 h with the indicated stimuli. (**E**) Time-course of *PCSK9* mRNA expression after incubation of hMφ with exogenous PCSK9. Values refer to HepG2 control condition. (**F**) Time-course of PCSK9 incorporation in hMφ polarized to M1-like or M2-like profiles. Resting (M0), M1, and M2 hMφ were treated with 0.5 μg/mL of recombinant human PCSK9, and the incorporation was determined by Western blot. Results show representative blots from 18 different donors (**A**,**B**), from 6 different donors and a representative image of PCSK9 incorporation (**C**–**E**), or 4 donors (**F**). * *p* < 0.05 vs. the corresponding M0 condition (**F**). F.I., fold induction (**D**,**E**).

**Figure 2 ijms-23-09114-f002:**
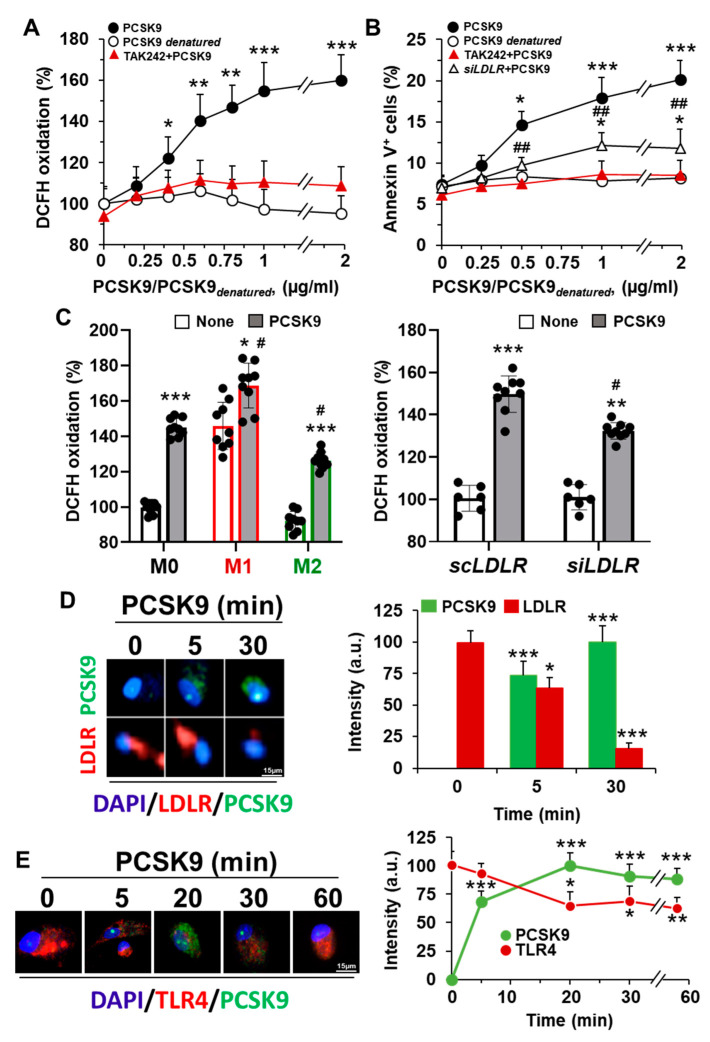
Treatment of human Mφ with PCSK9 enhances ROS synthesis and induces apoptosis. (**A**) PCSK9 dose-dependent oxidation of DCFH in hMφ treated for 18 h with recombinant PCSK9 or heat-inactivated PCSK9 (PCSK9_denatured_). When TLR4 activity was inhibited, cells were treated for 30 min with 1 μM TAK242 before PCSK9 incubation. (**B**) Apoptosis was evaluated by flow cytometry, quantifying the annexin V^+^ cells. Silencing of *LDLR* was done as described in the methods Section 4.6; (**C**) ROS production was determined in M0, M1-like, and M2-like polarized hMφ treated with the indicated stimuli 18 h before incubation with 1 μg/mL of PCSK9 for an additional period of 18 h. When *LDLR* was silenced, the time of incubation with *scRNA* or *siRNA* was 48 h before PCSK9 addition to the cell culture. (**D**,**E**) Representative immunofluorescence images of hMφ treated with 1 μg/mL of human recombinant PCSK9. At the indicated times, cells were fixed and stained with anti-PCSK9 antibody (in green), DAPI (in blue), and an anti-human LDLR antibody (**D**) or anti-human TLR4 antibody (both in red) (**E**). Data show the mean ± SD of 4 independent experiments. * *p* < 0.05, ** *p* < 0.005, and *** *p* < 0.001 vs. the control condition (**A**,**B**), vs. the corresponding vehicle condition for TAK242 (**C**), and vs. the value at time 0 (**D**,**E**), respectively. # *p* < 0.05 and ## *p* < 0.005 vs. the corresponding *siLDLR* condition and vs. the M0 or the *scLDLR* condition (**C**), respectively.

**Figure 3 ijms-23-09114-f003:**
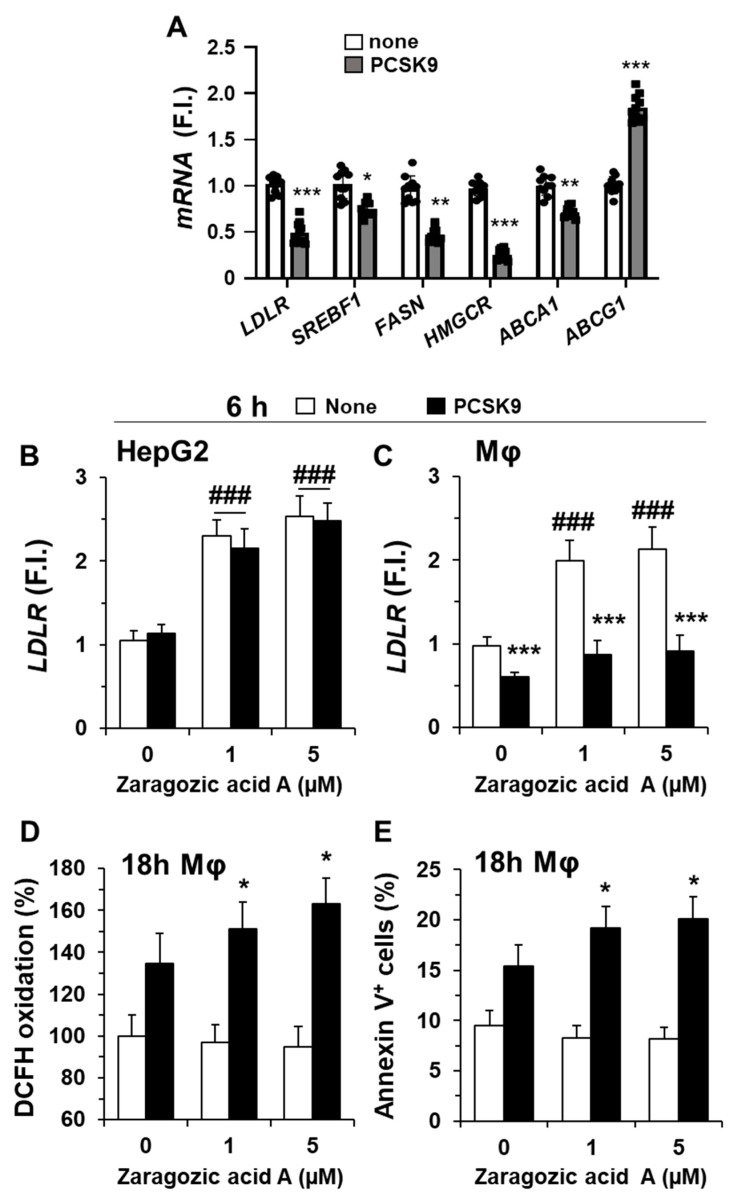
Incubation of human macrophages with PCSK9 reduces the expression of genes involved in lipid and cholesterol metabolism. (**A**) Quantification of *LDLR, SREBF1, FASN, HMGCR, ABCA1*, and *ABCG1* mRNA levels in hMφ after treatment with 0.5 μg/mL of PCSK9. Graphs represent the mean ± SD as the fold induction (F.I.) of each experimental group using *RPLP0* (*36B4*) expression as endogenous control. (**B**,**C**) HepG2 cells and hMφ were incubated for 24 h with the indicated concentrations of zaragozic acid A, followed by the addition of 0.5 μg/mL of PCSK9, and the *LDLR* mRNA levels were quantified. (**D**,**E**) The synthesis of ROS and the quantification of annexin V^+^ cells were determined after 18 h of incubation with PCSK9 in hMφ treated as described in panel C. Data show the mean ± SD of 6 independent donors (**A**) or from 4 independent experiments (**B**–**E**). * *p* < 0.05, ** *p* < 0.01, and *** *p* < 0.001 vs. the corresponding condition in the absence of PCSK9; ### *p* < 0.001 vs. the corresponding condition in the absence of zaragozic acid A.

**Figure 4 ijms-23-09114-f004:**
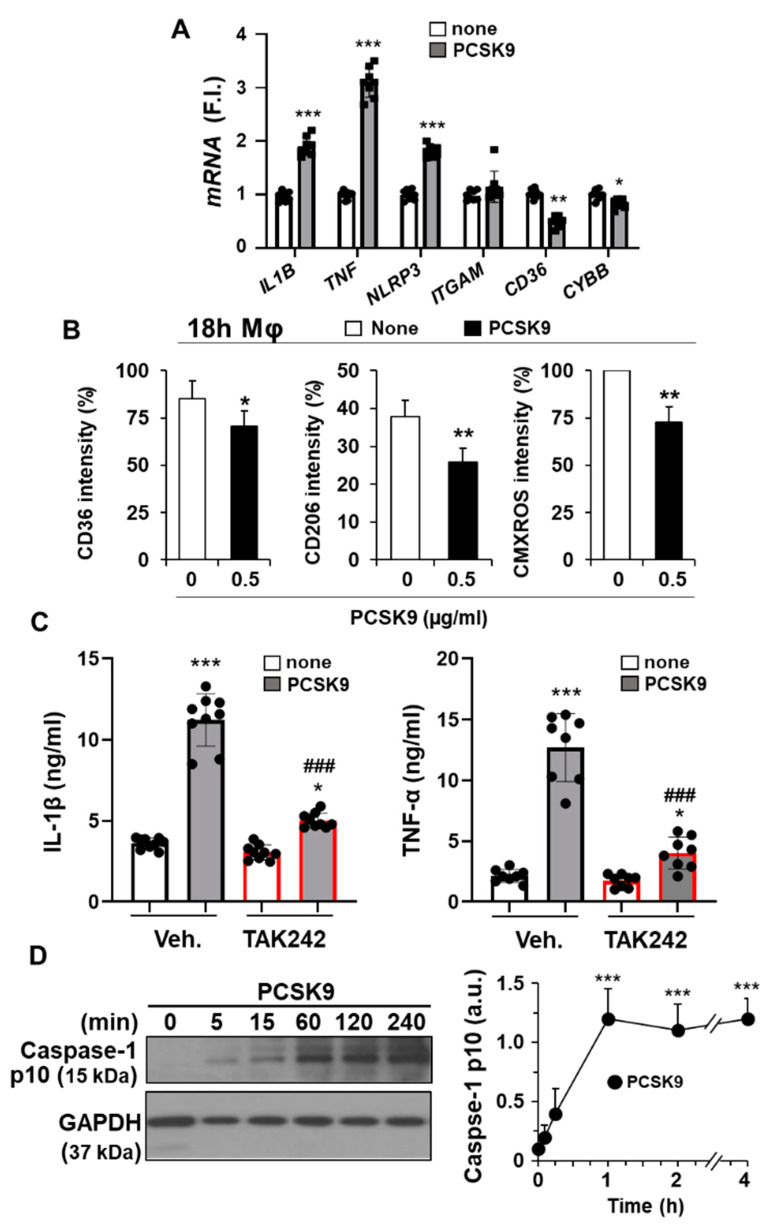
PCSK9 induces pro-inflammatory activation in human macrophages. (**A**) Quantification of mRNA levels of the pro-inflammatory mediators *IL1B, TNF, NLRP3, ITGAM, CD36*, and *CYBB* after treatment of hMφ with 0.5 μg/mL PCSK9. Graphs represent the mean ± SD as the fold induction (F.I.) of each experimental group using *RPLP0* expression as endogenous control. (**B**) Effect of the treatment of hMφ with 0.5 μg/mL PCSK9 on the cell surface levels of CD36 and CD206 and on the mitochondrial inner membrane potential, measured by the fluorescence of CMXROS. (**C**) IL-1β and TNF-α accumulation in the culture medium of hMφ after 4 h of incubation with 0.5 μg/mL of PCSK9. TAK242 (1 μM) was added 30 min before PCSK9 incubation. (**D**) Time-course of caspase-1 processing after treatment of hMφ with PCSK9. Data show the mean ± SD of 6 independent donors (**A**,**B**,**D**) or a representative blot out of 4 independent experiments (**C**). * *p* < 0.05, ** *p* < 0.01, and *** *p* < 0.001 vs. the corresponding condition in the absence of PCSK9; ### *p* < 0.001 vs. the corresponding vehicle condition for TAK242 (**C**,**D**).

**Figure 5 ijms-23-09114-f005:**
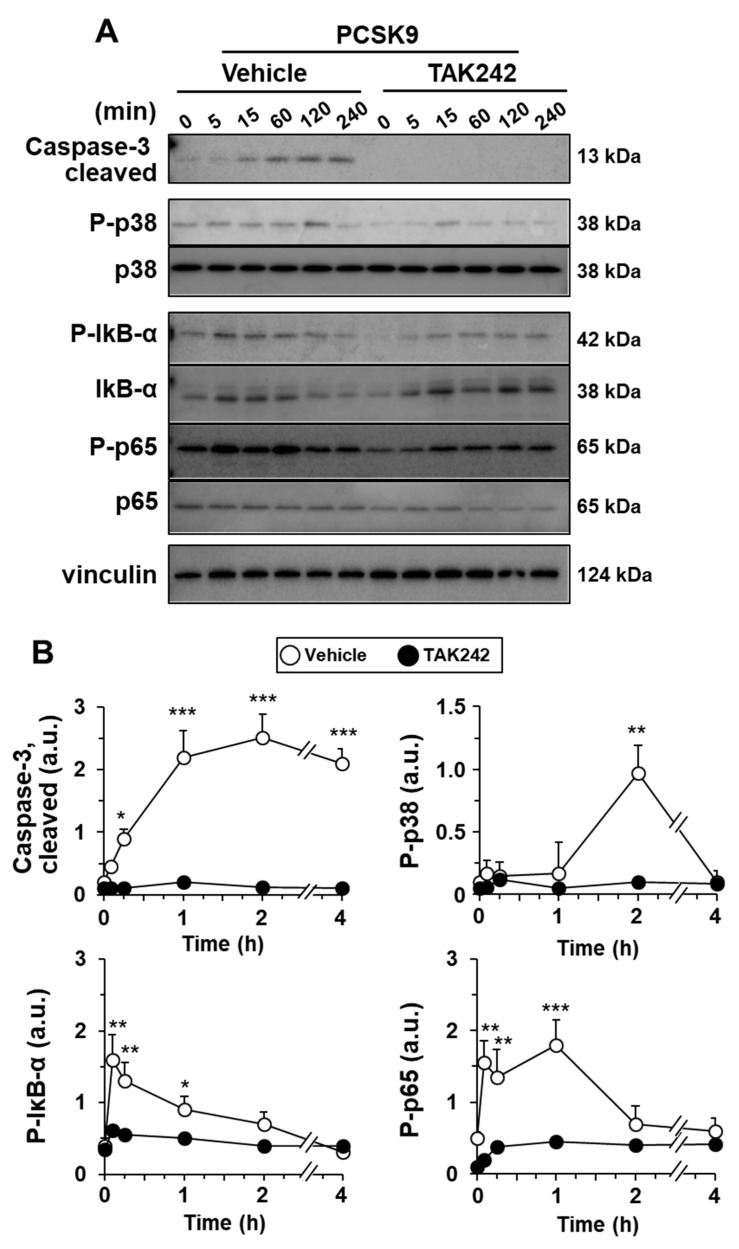
PCSK9 activates NF-κB, p38 MAPK, and caspase-3 processing in human macrophages. Cells were pre-treated for 30 min with vehicle or 1 μM of the TLR4 inhibitor TAK242 before incubation with 0.5 μg/mL of PCSK9. The time-course of the activation of caspase-3 and the phosphorylation of IκB-α, NF-κB protein p65, and the MAPK p38 were determined by Western blot (**A**); and the intensities of the bands were quantified (**B**). Results show representative blots and the quantification of 4 independent experiments (mean ± SD). * *p* < 0.05, ** *p* < 0.01, and *** *p* < 0.001 vs. the corresponding condition in the presence of TAK242.

## Data Availability

The data generated and analyzed during the current study are available upon request to the authors.

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
