# Peer review of "Functional Crosstalk between PCSK9 Internalization and Pro-Inflammatory Activation in Human Macrophages: Role of Reactive Oxygen Species Release"

_ijms, 2022, doi:10.3390/ijms23169114_

Round 1
Reviewer 1 Report
In this manuscript the authors describe downstream signalling events of PCSK9 in macrophages. In line with previous observations they found an uptake of PCSK9 into macrophages, together with LDL receptor degradation. Additionally, they suggest a link to TLR4 activation resulting in enhanced ROS production and induction of an inflammatory profile. Furthermore, they showed PCSK9 induced down regulation of several genes involved in lipid metabolism.
I have the following questions:
- How is PCSK9 taken up in LDLR KO/KD cells? And in TLR4 KO cells? Is there still PCSK9 uptake in double knockout cells?
- Do the authors think that LDLR and TLR4 are linked and activated together by PCSK9 (within one pathway)? Or do these receptors trigger individual signalling events? It would be of interest to analyse the ROS production and inflammatory response in a LDLR and TLR4 double knockout system (or knockdown / inhibitor treatment). This would help to establish a clear link between both receptors. Also is there any co-localisation of both receptors when PCSK9 is applied to the cells?
- PCSK9 seems to localize into the nucleus (Fig. 1), so do the authors suggest that PCSK9 by itself is regulating gene expression or via other transcription factors?
Minor comments:
- Fig. 1C scale bar is missing. Is it possible to increase the image size to improve the visability ?
- line 180 - 182: reference (s) missing
- Fig. 4B: representative images are missing or is this FACS staining (if so where are the FACS graphs)? How is the CD36 and CD206 plasma membrane staining intensity measured?
- Fig. 5 please add subpanel numbering
- line 300 and 324 references missing
- line 442: where are the FACS graphs/plots
Author Response
In this manuscript the authors describe downstream signalling events of PCSK9 in macrophages. In line with previous observations they found an uptake of PCSK9 into macrophages, together with LDL receptor degradation. Additionally, they suggest a link to TLR4 activation resulting in enhanced ROS production and induction of an inflammatory profile. Furthermore, they showed PCSK9 induced down regulation of several genes involved in lipid metabolism.
Firstly, we sincerely thank the Editor-in-chief and the Reviewers for their constructive comments and suggestions and the overall positive evaluation of our work. We have thoroughly revised the manuscript taking into account all the concerns and points raised by the Reviewers, and we now believe that the manuscript has improved considerably.
I have the following questions:
How is PCSK9 taken up in LDLR KO/KD cells? And in TLR4 KO cells? Is there still PCSK9 uptake in double knockout cells?
Do the authors think that LDLR and TLR4 are linked and activated together by PCSK9 (within one pathway)? Or do these receptors trigger individual signalling events? It would be of interest to analyse the ROS production and inflammatory response in a LDLR and TLR4 double knockout system (or knockdown / inhibitor treatment). This would help to establish a clear link between both receptors. Also is there any co-localisation of both receptors when PCSK9 is applied to the cells?
PCSK9 seems to localize into the nucleus (Fig. 1), so do the authors suggest that PCSK9 by itself is regulating gene expression or via other transcription factors?
We thank the Reviewer for this interesting point. As early as 2014, in Lagace's review of PCSK9 and LDLR degradation (Current Opinion in Lipidology; 25:387-393), this LDLR-independent uptake of PCSK9 is reported. We have observed that the incorporation kinetics of PCSK9 in LDLR KO murine macrophages is completely different and takes longer to incorporate. Although we have not investigated the process in depth, we think that the phagocytic activity of macrophages is more than sufficient to explain this process. In addition to this, the role of scavenger receptors (CD36, SRBI) in this process cannot be excluded and, indeed, the interaction of PCSK9 with CD36 has been described in various cell types, including macrophages. Furthermore, and as we discussed in the manuscript (ref. 60), it has been proposed that PCSK9 shares a resistin-like domain, which may explain TLR4 activation (resistin is a TLR4 agonist, as described in several reports; for example: Atherosclerosis. 2017; 259: 51–59). We do not have the LDLR/TLR4 DKO, but due to the suppressive effects of TLR4 inhibition with TAK242, we can speculate that this 'pro-inflammatory' phenotype, which is associated with PCSK9 uptake by human macrophages, will be attenuated or suppressed under these conditions.
Regarding the interaction between LDLR and TLR4 in macrophages, this is an open area. Crosstalk between TLR4/2 and minimally modified LDLR has been reported by at least several groups. In fact, when we analyzed the incorporation of PCSK9 by human macrophages and traced the dynamics of LDLR and TLR4 in parallel (Fig. 2D–E), the internalization kinetics of both receptors was very similar. We are working on this topic because we consider it highly relevant to understanding the role of PCSK9 in the fate of human macrophages and the progression of atherosclerosis; however, we prefer to use human macrophages rather than the genetically modified murine counterparts because, in the end, translation to human pathology is easier with human cells.
Regarding ROS production, and due to the potent inhibitory effect of TAK242 treatment, we think that the main contributor to the increase in ROS is the activation of TLR4/NF-kB. We have preliminary data targeting downstream steps in the TLR4/NF-kB pathway (IKK inhibition, etc.) that support this view. We are interested in identifying the steps responsible for this increase in ROS production because, in our opinion, this is highly relevant to the increase in oxLDL and the aggravation of the atherogenic process.
The Reviewer's observation of the presence of PCSK9 in the nucleus has been extensively reviewed. Although we cannot exclude the presence of PCSK9 in the nucleus at late periods, DAPI and PCSK9 co-staining are minimal at 30 min. Accumulation of PCSK9 is mainly in early endosomes. We are studying this process because it is possible that in subsequent periods the protein can be mobilized to other compartments, including the nucleus, but as shown in Fig. 1C, the accumulation is mainly in the cytoplasmic organelles (endosomes?). We thank the Reviewer for this observation and, following his/her recommendation, we have enlarged the images in Fig. 1C to clarify this point. Once again, the fate of PCSK9 in macrophages remains an interesting point of investigation.
Minor comments:
Fig. 1C scale bar is missing. Is it possible to increase the image size to improve the visability ?
Following the suggestion of the Reviewer we increased, as much as possible, the size of the right panel of Fig. 1C. Also, the size bars have been included in the figure.
line 180 - 182: reference (s) missing
Thank you for the correction. The references were corrupted when integrated with the IJMS platform and have been corrected.
Fig. 4B: representative images are missing or is this FACS staining (if so where are the FACS graphs)? How is the CD36 and CD206 plasma membrane staining intensity measured?
We use routinely FACS analysis of CD36 and CD206 in viable cells. This is described in the Methods section and details can be provided to readers upon request. In our opinion, this quantification is appropriate to provide the information we are trying to communicate to the readers. In addition, we are providing representative flow cytometry plots in the supplementary material.
Fig. 5 please add subpanel numbering
The requested numbering has been included. We thank the Reviewer for this observation since the definition of the symbols in the graphs was not correct.
line 300 and 324 references missing
Thank you. This has been corrected in the revised version.
line 442: where are the FACS graphs/plots
Thank you for this comment. The FACS analysis has been improved in the revised version, as well as representative flow cytometry plots that are included in the supplementary material.
Reviewer 2 Report
Rafael I Jaén et al have studied the effect of PCSK9 on the inflammatory activation of human macrophages. They have shown that macrophages rapidly take up PCSK9 and activate proinflammatory cytokine production through a TLR4-dependent pathway.
PCSK9 is one of the main therapeutic targets for cardiovascular prevention and therapy; therefore, the new insights into the effects of PCSK9 are extremely timely and interesting.
This article contains a great deal of detailed information on the mechanisms of inflammation associated with the action of PCSK9. The study design is adequate, the methods are sound, and the results and conclusions are highly reliable.
I have the following comments that I would like to see addressed by the authors:
1. The authors conclude that macrophages do not express PCSK9, and it is internalized against media conditioned with HepG2 (Fig1C) or recombinant PCSK9 (Fig1E). These data show an increase in intracellular levels of PCSK9 protein, but do not rule out that media conditioned with recombinant PCSK9 or HepG2 can induce endogenous expression (mRNA levels) of PCSK9. Therefore, the authors should moderate/tone down this conclusion by at least demonstrating that cells challenged with recombinant PCSK9/HepG2 conditioned media do not induce expression of endogenous PCSK9.
2. In Fig. 2A and B, cells treated with recombinant PCSK9 are indicated as "vehicle" (closed circles) compared to cells receiving the TLR4 inhibitor TAK242 (red triangles). Although indicated in the figure caption, it is suggested that in the figure the back circles be indicated as rPCSK9 instead of "vehicle" and the red triangles as rPCSK9+TAK242. Following the same criteria, the white circles should be indicated as dPCSK9 (PCSK9 denatured).
3. In Fig2B, cells transfected with siLDLR (open triangles) should be compared with cells transfected with a scRNA, as shown in Fig2C, right. The percentage of LDLR silencing using siRNA should be shown.
4. What is the mechanism through which PCSK9 decreases the expression (mRNA levels) of LDLR (Fig3A)? Similarly, how does PCSK9 reduce the expression of SRBF1?
5. LDLR silencing reduced PCSK9-induced ROS production (Fig. 2C) and apoptosis (Fig. 2B). However, reduction of LDLR by PCSK9 in cells challenged with zaragozic acid A (Fig. 3C) increases ROS production (Fig. 3D) and apoptosis (Fig. 3E). Are the effects of PCSK9 regulating ROS and apoptosis dependent on LDLR or not?
6. In Figure 5, open circles refer to cells exposed to PCSK9+TAK242, while black circles refer to cells exposed to PCSK9+vehicle. This must be correctly indicated in the legend of the figure.
In the discussion, the authors focus their attention on the effects that could be mediated by SREBP-1, however, the main transcription factor associated with PCSK9 and cholesterol synthesis (HMGCoAr) and cell acquisition (LDLR) is SREBP-2 . Why don't you comment on this transcription factor? This was very surprising to me.
The comments on statins and PCSK9 on page 14 are at least inaccurate and require some amendment:
· Line 353: PCSK9 inhibitors are used in patients with HF and patients with established atherosclerotic cardiovascular disease (eliminate revascularization that is included in the ASCVD concept).
· Line 354: PCSK9 inhibitors are not effective in lowering triglycerides.
· Line 355: The impact on ASCVD has already been mentioned.
· Line 356-7: Subcutaneous administration is not a disadvantage.
· Line 358: PCSK9 inhibitors are not an alternative option to statins but a complementary therapy.
· Line 360: The efficacy of statins is not questioned. Its safety and effectiveness are backed by tons of scientific evidence. Please remove this incorrect and dangerous sentence.
· Line 362: Statins lower cholesterol and also have so-called pleiotropic effects, including a strong anti-inflammatory effect. The literature is abundant in data on the anti-inflammatory effects of statins.
I suggest that a clinician review the clinical implications of this article, included in the discussion section.
Author Response
Rafael I Jaén et al have studied the effect of PCSK9 on the inflammatory activation of human macrophages. They have shown that macrophages rapidly take up PCSK9 and activate proinflammatory cytokine production through a TLR4-dependent pathway.
PCSK9 is one of the main therapeutic targets for cardiovascular prevention and therapy; therefore, the new insights into the effects of PCSK9 are extremely timely and interesting.
This article contains a great deal of detailed information on the mechanisms of inflammation associated with the action of PCSK9. The study design is adequate, the methods are sound, and the results and conclusions are highly reliable.
We sincerely thank the Reviewer for the constructive comments and suggestions and the overall positive evaluation of our work. We have thoroughly revised the manuscript taking into account all the concerns and points raised by the two Reviewers, and we now believe that the manuscript has improved considerably.
I have the following comments that I would like to see addressed by the authors:
The authors conclude that macrophages do not express PCSK9, and it is internalized against media conditioned with HepG2 (Fig1C) or recombinant PCSK9 (Fig1E). These data show an increase in intracellular levels of PCSK9 protein, but do not rule out that media conditioned with recombinant PCSK9 or HepG2 can induce endogenous expression (mRNA levels) of PCSK9. Therefore, the authors should moderate/tone down this conclusion by at least demonstrating that cells challenged with recombinant PCSK9/HepG2 conditioned media do not induce expression of endogenous PCSK9.
We thank the Reviewer for this interesting comment. We performed experiments with human macrophages incubated with recombinant PCSK9 and quantified PCSK9 mRNA levels up to 48 hours. We used cells from different donors and the Ct values were very high ​​and the quantification of possible gene expression was negative. Since these data were available, they have been included in the new version (Fig. 1D). Furthermore, we know that M1/M2 polarization did not alter this negative result. However, the possibility that HepG2 cell-culture supernatants may induce an increase in gene transcription cannot be ruled out.
In Fig. 2A and B, cells treated with recombinant PCSK9 are indicated as "vehicle" (closed circles) compared to cells receiving the TLR4 inhibitor TAK242 (red triangles). Although indicated in the figure caption, it is suggested that in the figure the back circles be indicated as rPCSK9 instead of "vehicle" and the red triangles as rPCSK9+TAK242. Following the same criteria, the white circles should be indicated as dPCSK9 (PCSK9 denatured).
We sincerely thank the Reviewer for this indication that simplifies how the results are presented. These changes have been included.
In Fig2B, cells transfected with siLDLR (open triangles) should be compared with cells transfected with a scRNA, as shown in Fig2C, right. The percentage of LDLR silencing using siRNA should be shown.
Thank you for this comment. In the methods section, we have added precise information regarding the levels of PCSK9 mRNA after silencing and the amount of protein; specifically the LDLR present in the plasma membrane, that was negligible. The antibody that we have against LDLR is of poor quality, but after comparing the scRNA vs the siRNA and the expected size of the protein we have deduced the extent of silencing that is described in the methods section.
What is the mechanism through which PCSK9 decreases the expression (mRNA levels) of LDLR (Fig3A)? Similarly, how does PCSK9 reduce the expression of SRBF1?
We are working on this transcriptional control of LDLR in macrophages incubated with PCSK9. The repression described in Fig. 3A has been observed in all the blood donors analyzed. We were surprised by this result since, in our opinion, a rise should be expected. However, because ROS are produced and it is expected that LXR and other nuclear receptors that repress LDLR transcription are activated, this is a potential way. We are continuing this specific study in human macrophages to understand the mechanisms involved in this transcriptional regulation. In the same aspect, the differential regulation of ABCA1 vs. ABCG1 was surprising and we need to segregate the LDLR- vs the TLR4-dependent pathways to dissect the transcriptional control and to identify the factors involved in these processes. The same applies to SREBF1, which also controls the transcription of different genes relevant to lipid metabolism, but also in functional polarization (Nat Metab. 2021; 3:1150–1162).
LDLR silencing reduced PCSK9-induced ROS production (Fig. 2C) and apoptosis (Fig. 2B). However, reduction of LDLR by PCSK9 in cells challenged with zaragozic acid A (Fig. 3C) increases ROS production (Fig. 3D) and apoptosis (Fig. 3E). Are the effects of PCSK9 regulating ROS and apoptosis dependent on LDLR or not?
We used zaragozic acid to enhance the levels of LDLR. This treatment did not alter the basal production of ROS in macrophages. We aimed to see if enhanced LDLR levels were associated with increased ROS production upon incubation of human macrophages with PCSK9. As Fig 3D and E show, these increased LDLR levels were accompanied by enhanced ROS production and apoptosis, supporting our view that LDLR levels are relevant to the PCSK9-dependent activation response in macrophages.
In Figure 5, open circles refer to cells exposed to PCSK9+TAK242, while black circles refer to cells exposed to PCSK9+vehicle. This must be correctly indicated in the legend of the figure.
We thank the Reviewer for this observation. The definition of the symbols has been maintained and the graphs have been corrected.
In the discussion, the authors focus their attention on the effects that could be mediated by SREBP-1, however, the main transcription factor associated with PCSK9 and cholesterol synthesis (HMGCoAr) and cell acquisition (LDLR) is SREBP-2. Why don't you comment on this transcription factor? This was very surprising to me.
We thank the Reviewer for this point that was discussed in the group. We agree with your comment on the relative roles of SREBP-1 and SREBP-2. Because SREBP-1 is under the control of SREBP-2 and in macrophages, SREBP-1 is regulating the expression of scavenger efferocytosis/phagocytosis receptors we considered focusing on this transcription factor. In fact, the primary target that we focused on was LXR regulation, and the tandem LXR/SREBP-1 is very important in the regulation of cholesterol metabolism and efflux in macrophages. In our opinion, SREBP-1 is integrating the actions of SREBP-2. However, and following your very interesting suggestion we think that, to continue the work, we need to include SREPB-2 in the analysis, because the drop in SREBP1 mRNA after PCSK9 incubation suggests that perhaps this is not a direct effect and is dependent on the fate of SREBP-2.
The comments on statins and PCSK9 on page 14 are at least inaccurate and require some amendment:
- Line 353: PCSK9 inhibitors are used in patients with HF and patients with established atherosclerotic cardiovascular disease (eliminate revascularization that is included in the ASCVD concept).
- Line 354: PCSK9 inhibitors are not effective in lowering triglycerides.
- Line 355: The impact on ASCVD has already been mentioned.
- Line 356-7: Subcutaneous administration is not a disadvantage.
- Line 358: PCSK9 inhibitors are not an alternative option to statins but a complementary therapy.
- Line 360: The efficacy of statins is not questioned. Its safety and effectiveness are backed by tons of scientific evidence. Please remove this incorrect and dangerous sentence.
- Line 362: Statins lower cholesterol and also have so-called pleiotropic effects, including a strong anti-inflammatory effect. The literature is abundant in data on the anti-inflammatory effects of statins.
I suggest that a clinician review the clinical implications of this article, included in the discussion section.
Many thanks for these comments, because our opinion on these aspects was not neutral! Following the indications of the Reviewer, we have modified the text in the indicated lines. The therapeutic indications for PCSK9 antibodies/inhibitors (when available!) are very specific and the use of statins constitutes a primary option. Accordingly, we have presented the use of PCSK9 inhibitors and statins neutrally, with the specific indications for each one.
We have also discussed these points with clinicians (internal medicine and cardiologists). A question that has not been investigated in depth is regarding the high doses of statins administered to patients after a heart infarction.
We hope that, upon revision, the positive aspects of both PCSK9 inhibitors and statins are correctly balanced.
Round 2
Reviewer 2 Report
The authors have correctly answered my questions and have adapted the text accordingly.
Please include a sentence in line with your comments to my question about SREBP1 and 2 in the discussion section of the text.
Please amend the sentence on page 19 Line 1 that says that statins reduce LDLR expression (!!!). The main pharmacodynamic effect of statins is the increase of LDLR.
No more comments.
Author Response
We thank the Reviewer for the comments addressed through the evaluation process that has improved the quality of the manuscript. In particular, we apologize for the error in the regulation of LDLR levels by statins, that was not corrected in the previous revision.
The authors have correctly answered my questions and have adapted the text accordingly.
Please include a sentence in line with your comments to my question about SREBP1 and 2 in the discussion section of the text.
Following your comment, the following sentences have been included in lines 297-305 (pdf format):
SREBP-1 is under the control of SREBP-2 and, in macrophages, SREBP-1 regulates the expression of scavenger receptors involved in efferocytosis and phagocytosis, linking lipid metabolism and inflammatory responses [53]. Indeed, crosstalk between the LXR and SREBP-1 transcription factors is relevant in the regulation of cholesterol metabolism and efflux in macrophages [52-53]. Therefore, the ability of PCSK9 to modulate SREBP-1 and its targets can contribute to lipid metabolism dysregulation [55]. Together, these results indicate that PCSK9 uptake causes an alteration in the lipid profile in hMφ that is involved in foam cell formation. This effect appears to be dependent on LDLR, as described in murine Mφ [1].
Please amend the sentence on page 19 Line 1 that says that statins reduce LDLR expression (!!!). The main pharmacodynamic effect of statins is the increase of LDLR.
We apologize for this error that was observed in the previous version but finally was not amended. The sentence now reads (line 359 pdf version):
In fact, statins increase the expression of the LDLR, although they favor the induction of the p38 and NLRP3 inflammasome pathways and increased PCSK9 levels.